# The Biological Effects of Ozone Gas on Soft and Hard Dental Tissues and the Impact on Human Gingival Fibroblasts and Gingival Keratinocytes

Alin Daniel Floare [1], Alexandra Denisa Scurtu [2,3,*], Octavia Iulia Balean [1,4,*], Doina Chioran [5], Roxana Buzatu [5], Ruxandra Sava Rosianu [1,4], Vlad Tiberiu Alexa [1], Daniela Jumanca [1,4], Laura-Cristina Rusu [1,6], Robert Cosmin Racea [1,6], Dorina Coricovac [2,3], Iulia Pinzaru [2,3], Cristina Adriana Dehelean [2,3] and Atena Galuscan [1,4]



1    Departament I, Faculty of Dental Medicine, "Victor Babes" University of Medicine and Pharmacy, Eftimie Murgu Square, No. 2, 300041 Timisoara, Romania; floare.alin@umft.ro (A.D.F.); sava-rosianu.ruxandra@umft.ro (R.S.R.); vlad.alexa@umft.ro (V.T.A.); jumanca.daniela@umft.ro (D.J.); laura.rusu@umft.ro (L.-C.R.); robert.racea@gmail.com (R.C.R.); galuscan.atena@umft.ro (A.G.)

2    Department of Toxicology, Faculty of Pharmacy, "Victor Babes" University of Medicine and Pharmacy, Eftimie Murgu Square, No. 2, 300041 Timisoara, Romania; dorinacoricovac@umft.ro (D.C.); iuliapinzaru@umft.ro (I.P.); cadehelean@umft.ro (C.A.D.)

3    Research Centre for Pharmaco-Toxicological Evaluation, "Victor Babes" University of Medicine and Pharmacy, Eftimie Murgu Square, No. 2, 300041 Timisoara, Romania

4    Translational and Experimental Clinical Research Center in Oral Health (TEXC-OH), 14A Tudor Vladimirescu Ave., 300173 Timisoara, Romania

5    Department II, Faculty of Dental Medicine, "Victor Babes" University of Medicine and Pharmacy, Eftimie Murgu Square, No. 2, 300041 Timisoara, Romania; caruntud@yahoo.com (D.C.); drbuzaturoxana@gmail.com (R.B.)

6    Multidisciplinary Center for Research, Evaluation, Diagnosis and Therapies in Oral Medicine, "Victor Babes" University of Medicine and Pharmacy, Eftimie Murgu Square, No. 2, 300041 Timisoara, Romania

*    Correspondence: alexandra.scurtu@umft.ro (A.D.S.); balean.octavia@umft.ro (O.I.B.)

**Abstract:** Ozone is an allotropic form of oxygen, so in the medical field ozone therapy has special effects. Starting from the premise that bio-oxidative ozone therapy reduces the number of bacteria, in the present study two approaches were proposed: to evaluate the biological effects of ozone gas on the tooth enamel remineralization process and to demonstrate its impact on the morphology and confluence of human primary gingival cells, namely keratinocytes (PGK) and fibroblasts (HGF). The ozone produced by HealOzone was applied in vivo to 68 M1s (first permanent molars), both maxillary and mandibular, on the occlusal surfaces at pit and fissure. The molars included in the study recorded values between 13 and 24 according to the DIAGNOdent Pen 2190 scale, this being the main inclusion/exclusion criterion for the investigated molars. Because the gas can make contact with primary gingival cells during the ozonation process, both human gingival fibroblasts and keratinocytes were exposed to different doses of ozone (20 s, 40 s, 60 s), and its effects were observed with the Olympus IX73 inverted microscope. The contact of ozone with the human primary gingival cells demonstrates cell sensitivity to the action of ozone, this being higher in fibroblasts compared to keratinocytes, but it is not considered toxic because all the changes are reversible at 48 h after exposure.

**Keywords:** ozone; dental remineralization; primary gingival fibroblasts; primary gingival keratinocytes



## 1. Introduction

Ozone ($O_3$) is a naturally occurring gaseous molecule that was discovered in the mid-nineteenth century [1]. It is an assembly of three oxygen atoms with a short half-life (40 min at 20 °C) and is highly unstable. In comparison with oxygen, ozone is denser (1.6-fold), has a greater solubility in water (10-fold), and is considered the third most potent oxidant (after

fluorine and sulfate) [2]. Its oxidant potential exceeds chloride (1.5 times greater) in terms of antimicrobial activity against viruses, bacteria, fungi, and protozoa [2,3]. Ozone acts on bacterial cell membranes by oxidizing lipid and lipoprotein cell components, causing the deterioration of the internal bacterial membrane. All viruses (including SARS-CoV-2) are ozone sensitive, especially those with a lipid coating. Viral component analysis shows the deterioration of polypeptide chains and envelope proteins, which greatly reduces the viral attachment capacity [4,5].

A plethora of proven health benefits were described for ozone, as follows: the stimulation of oxygen metabolism; an increase in the glycolysis rate of red blood cells and activation of the immune system; the stimulation of immunoglobulin synthesis and proliferation in immune cells; an anti-inflammatory effect by promoting the synthesis of interleukins, prostaglandins, and leukotrienes; and an augmented wound healing process [6,7].

In the view of all of ozone's therapeutic effects stated above, the mechanisms of action are also diversified, such as: (i) as an antimicrobial agent, $O_3$ destroys viral capsid and disrupts the reproductive cycle via peroxidation; (ii) as an immunomodulatory and antioxidant agent, it stimulates immunoglobulin synthesis, preserves cell redox state, and increases glutathione (GSH) peroxidase cellular levels; (iii) the anti-hypoxic effect occurs by enhancing oxygen saturation of hemoglobin; (iv) the anti-inflammatory action is due to the impact of ozone in interleukins, prostaglandins, and leukotrienes synthesis; (v) the bioenergetic effect is exerted via the activation of protein synthesis and an enhancement of cellular metabolism; (vi) the detoxification activity occurs by activating the cellular aerobic processes (Krebs cycle, glycolysis and fatty acids oxidation); and (vii) the biosynthetic function is realized by enhancing the metabolism of carbohydrates, proteins, and lipids [8–12].

Other than the multiple uses of ozone therapy in the medical domain (ocular pathologies, acute and chronic bacterial and viral infections, orthopedic disorders, renal, pulmonary, hematological and neurodegenerative pathologies), in recent years, it has become of great interest in the application of ozone therapy in dentistry [3,11,13].

Regarding oral infections (canker sores, opportunistic infections, or acne), the application of ozone as quickly as possible leads to a reduced multiplication of the pathogen, improved microcirculation, reduced inflammation, and the achieving of a rapid cure [14].

Ozone can be administered as gaseous form topically using an open system or a sealing suction one, as ozonated water or as ozonized oil (using sunflower oil—commercial names Oleozone, Bioperoxoil) [12,15]. In the dental field, ozone has been advocated for use as a treatment of gum infections, during surgery, for failed implant cases, root caries, and root canal treatment [16].

Several studies have evaluated the effect of ozone use on cavities, pits, fissures, and primary root cavities, and found that non-cavity lesions treated with ozone proved susceptible to remineralization [17–22]. The use of ozone did not influence the physical properties of the enamel, either by increasing or by reducing the sealing capacity. Furthermore, it is of note that ozone can be used both on the intact enamel and during the remineralization process of the initial and medium primary cavities.

Celiberti et al. reached the conclusion that the application of ozone does not affect the modulus of elasticity and hardness of dentin. Thus, ozone could be applied to dentine without affecting the micro-mechanical properties of the substrate [23].

In recent years, researchers have shown considerable interest in the antibacterial properties of ozone. Several studies have been carried out, which support the benefits of ozone therapy and its usefulness in reducing the number of bacteria from different surfaces, such as dental tissue or acrylic denture plates [24–27].

The biological action of ozone gas on bacteria on dental surfaces must also be supported by the effect on the soft gingival tissues that may come into contact with the gas during dental therapy. Therefore, the cytotoxic effects of ozone gas are important, as they are regulated by organizations such as the National Institute of Clinical Excellence (NICE). As there is limited evidence on the benefits of gaseous ozone in its application in



dentistry, current National Institute of Clinical Excellence guidelines advise against using ozone alone in the treatment of caries in general dental practice. Millar and Hodson have evaluated the safety of two ozone delivery devices designed for use in dentistry. OziCure and HealOzone devices were used in a clinical simulation using a phantom head, whereas recordings of ozone levels were made in the pharyngeal and nasal regions of the patient and near the mouth of the operator. According to the results, the authors concluded that the HealOzone (KaVo CO, Biberach/Riss, Germany, GmbH) device was safe to use [28–30].

Even though ozone exerts multiple oral health beneficial effects, it must be remembered that ozone is a strong oxidizing compound and can be very toxic for the bronchio-pulmonary system after inhalation, causing a sore throat, augmentation of asthma, and even lung damage [31]. Long-term exposure to ozone is associated with several side effects, such as irritation of the upper airways, epiphora, rhinitis, coughing, bronchoconstriction, headaches, and vomiting [7].

Starting from the premise that bio-oxidative ozone therapy reduces the number of bacteria, this study aims to evaluate the biological effects of ozone gas on the tooth enamel remineralization process and to verify the impact of gaseous ozone therapy on the morphology and confluence of human primary gingival cells: keratinocytes (PGK) and fibroblasts (HGF), which represent most of the oral cavity resident cells.

## 2. Materials and Methods

The in vivo protocol for this study was approved by the Commission of Ethics for Scientific Research of the "Victor Babes" University of Medicine and Pharmacy.

### 2.1. Part One of the Experiment

The experimental part 1 assessed the benefits of bio-oxidative therapy on occlusal pits and fissures in the first permanent molars (M1s) through the ozonation process.

From the 44 patients selected for the study, aged 6 to 12, 88 occlusal surfaces of maxillary and mandibular molars met the criteria for inclusion.

Out of the selected patients, 38 agreed to take part in the study. Hereby, the dental surfaces of all molars were separately examined, according to the inclusion/exclusion criteria, leaving us with a total of 68 molars (M1s) for research.

2.1.1. Study Design

1. M1s evaluation by laser fluorescence (DIAGNOdent Pen 2190) to identify enamel quality and initial demineralization; 2. Ozonation; 3. Immediate re-evaluation; 4. Re-evaluation one month after the ozonation.

Only fully erupted molars and integral molars without clinically visible cavitary lesions, recorded according to the evaluation scale with values between 13 and 24 (<25), were included in the study, this being the main inclusion/exclusion criterion for the investigated molars. The maximum admissible value for the laser fluorescence device recording was up to 24, which is equivalent to enamel with mild demineralization, with a medium risk of dentinal carious lesion. The chosen teeth had no or slight modification in enamel translucency after prolonged air drying (5 s), scored 0 according to ICDAS-II, and were diagnosed by laser fluorescence showing a reading of less than 24, indicating sound enamel [32].

The teeth selection criteria considered the absence of dentin or cavitary lesions so that measurements after ozonation were as sensitive and reproducible as possible with minimal errors. After a professional cleaning, the teeth that were to be included in the study were dried and isolated. For the correct use of DIAGNOdent Pen 2190, the equipment was first calibrated for each patient, because each individual showed varying degrees of dental mineralization. Calibration was performed on an appropriate dental surface on the front teeth. As a calibration tooth, we used teeth 1.1 or 2.1, the central permanent incisors. After calibration, the peak of the laser fluorescence value was recorded for each occlusal surface after three readings. This allowed us to determine a value for each occlusal molar surface.

Once the mineralization values of the teeth were selected and measured, we proceeded to the next stage, namely the application of ozone using HealOzone, the X4 model (KaVo Biberach/Riss, Germany).

### 2.1.2. Ozonation Procedure

The ozonation procedure consists of a package that includes: The application of ozone gas, the use of remineralizing agents, and a patient kit and information on oral hygiene. The ozonation device comprises an air filter, vacuum pump, an ozone generator, a handpiece fitted with a sealing silicone cup, and a flexible hose. The procedure usually takes between 20 and 120 s per tooth. According to the specifications, the device produces ozone at a concentration of 32 g/m$^3$ at an exposure of 60 s. For all dental units, the 40 s contact time was chosen, this being the optimal ozone treatment interval [30,33–35]. The ozonation phase was followed by a second measurement using laser fluorescence, during the same session, this time to evaluate the changes in the dental enamel in grooves and fossa. After one month, the patients were re-evaluated with a new measurement with the same equipment and using the same calibration parameters to see if any changes occurred in the ozone-treated teeth after 30 days.

### 2.2. Part Two of the Experiment

### 2.2.1. Cell Lines

To perform this experiment, the following cell lines were purchased from ATCC (American Type Culture Collection): primary gingival keratinocytes (PGK—ATCC® PCS-200-014™) and primary gingival fibroblasts (HGF—ATCC® PCS-201-018™).

Keratinocytes present an epithelial morphology, are adherent, have a round shape, and when they reach a confluence close to 80%, they form colonies. These cells represent an experimental model with multiple applications in scientific research of the oral cavity, being applied in studies on antibiotic treatment and to verify the body's response to various dental implants, but their area of application is not limited to only to these kinds of studies [36].

Primary gingival fibroblasts of human origin—HGF are adherent, spindle-shaped and bipolar cells. This type of cell can be used as an alternative source of mesenchymal stem cells due to its similar morphology [37].

### 2.2.2. Cell Culture

Culture and growth of primary human gingival keratinocytes require a specific culture medium: Dermal Cell Basal Medium (ATCC® PCS-200-030™) that has been enriched with a specific growth kit, namely the Keratinocyte Growth Kit (ATCC® PCS- 200-040™). In the case of fibroblasts, we used a different culture medium: Fibroblast Basal Medium (ATCC PCS-201-030) that was supplemented with its own growth kit, the Fibroblast Growth Kit-Low Serum (ATCC PCS-201-041). This kit contains the following components: L- glutamine; ascorbic acid; hemisuccinate hydrocortisone; fetal bovine serum; insulin obtained by recombinant DNA technique; and recombinant fibroblast growth factor (FGF). Throughout the experiment, the cells were maintained under standard conditions: in an incubator at 37 °C in an atmosphere composed of 95% O$^2$ and 5% CO$^2$. Cell splitting was performed when they reached the appropriate confluence (85–90%) after 48 to 72 h in culture.

### 2.2.3. Experimental Design

In order to observe the possible changes at the cells' morphology level and confluence induced by gaseous ozone therapy, several approaches were proposed, such as:

Single exposure for 20 s of both primary gingival fibroblasts and keratinocytes;
Single exposure for 40 s of primary gingival keratinocytes;
Single exposure for 60 s of primary gingival keratinocytes.

Time for the exposure was selected according to exposure needed during dental treatments. The gaseous ozone was obtained with the ozonation device (HealOzone X4) at a concentration of 32 g/m$^3$ and an exposure time of 60 s [35].

2.2.4. Impact on Cell Morphology and Confluence

An Olympus IX73 inverted microscope equipped with a DP74 camera was used to observe the changes induced by ozone therapy at the level of cell morphology and confluence, and the acquisition and processing of images was performed using CellSens V1.15 software (Olympus, Tokyo, Japan). The pictures were taken at objectives 4, 10, and 20× after 24 and 48 h post-exposure to ozone gas.

## 3. Results

### 3.1. Experiment Part ONE

Measurements made using laser fluorescence were recorded and entered into a Microsoft Office Excel table. These measurements were statistically analyzed using IBM-SPSS 18, 2010. The threshold value for statistical significance was set at $p < 0.05$. The probability value $p$ was recorded for each test applied, with 2 decimals. For the comparisons of measured data, we applied the ANOVA test for repeated design.

Evolution of the Degree of Mineralization of All the Studied Teeth

Using the mean values per patient calculated for the molars, we applied the ANOVA test for repeated design to see whether there were differences in the tooth mineralization degree during the intervention. Mineralization is significantly different between determinations F (1.9) = 55.71, $p < 0.05$ (Figure 1). After further investigation, statistically significant differences were determined between the initial and post-ozone assessment ($p < 0.05$), as well as between the post-ozone and the one-month findings ($p < 0.05$). Between the initial and post-ozone conditions, the difference between the degree of mineralization was m = 6.37 and the standard error = 1.03, $p < 0.05$, which is statistically significant. Between post-ozone and one-month after the application of ozone conditions, the difference between the degree of mineralization was m = 1.77 and the standard error = 2.51, $p < 0.05$, which is also statistically significant (Table 1).

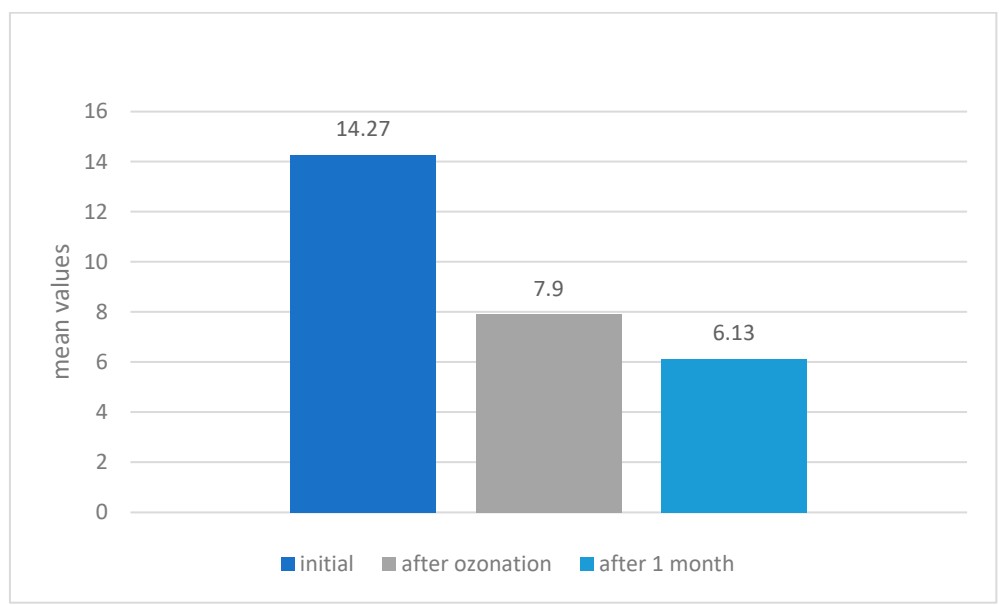

**Figure 1.** Evolution of the remineralization degree of all examined teeth (initial, after ozonation, 1 month after).

**Table 1.** ANOVA test results—mineralization of all examined teeth. (* $p < 0.05$).

| Evaluation Stage | | Mean Difference | Std. Error | 95% Confidence Interval for Difference | |
|---|---|---|---|---|---|
| | | | | Lower Bound | Upper Bound |
| Initial stage | immediate post-ozone | 6.375 * | 1.032 | 3.358 | 9.392 |
| | 1 month post-ozone | 8.150 * | 1.060 | 5.051 | 11.249 |
| Immediate post-ozone | initial | −6.375 * | 1.032 | −9.392 | −3.358 |
| | 1 month post-ozone | 1.775 * | 0.251 | 1.041 | 2.509 |
| 1 month post-ozone | initial | −8.150 * | 1.060 | −11.249 | −5.051 |
| | immediate post-ozone | −1.775 * | 0.251 | −2.509 | −1.041 |

### 3.2. Experiment Part Two

3.2.1. Gaseous Ozone Affects Primary Gingival Fibroblasts' Morphology and Confluence

A single exposure to gaseous ozone for 20 s induced significant changes in the fibroblasts' morphology and confluence at both 24 and 48 h post-exposure (see Figures 2 and 3), which were as follows: confluence was very low at 24 h after exposure, and the cells, although still adherent to the plate, displayed a different shape (were much more elongated and had some kind of vesicles on the surface; a greater quantity of debris was observed in the culture medium due to cell degradation) compared to that observed in control cells not exposed to ozone (Figure 2).

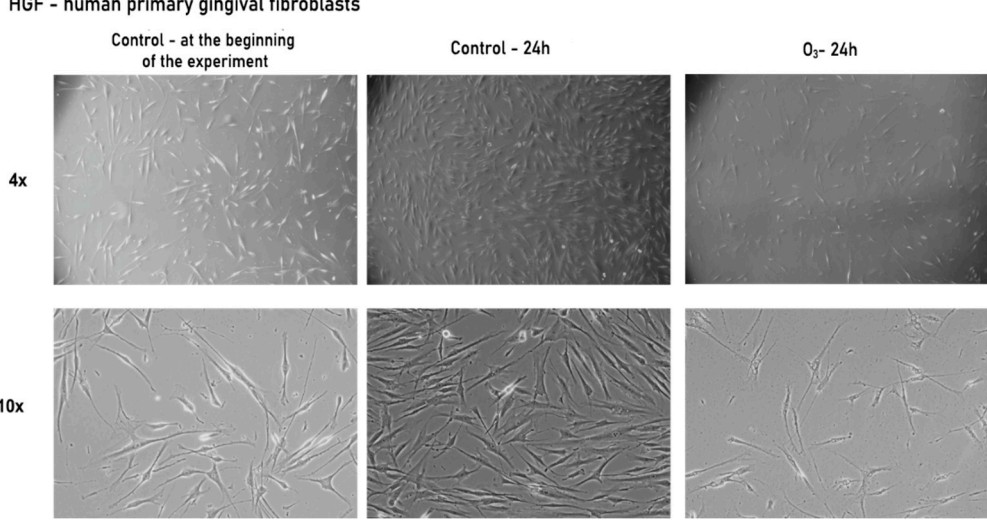

**Figure 2.** Microscopical aspect of the primary gingival fibroblasts—HGF in culture at 24 h post-exposure to ozone gas for 20 s. Pictures were taken with 4× and 10× lenses.

At 48 h post-exposure to ozone, fibroblasts began to regain their initial morphology, similar to the one exhibited by the control cells. An increase in confluence was also observed compared to those from 24 h exposure, but was still lower than that of control cells (see Figure 3).

**HGF – human primary gingival fibroblasts**

Control – 48h          O₃–48h

4x

10x

**Figure 3.** Microscopical aspect of the primary gingival fibroblasts—HGF in culture at 48 h post-exposure to ozone gas for 20 s. Pictures were taken with 4× and 10× lenses.

3.2.2. Effect of Ozone Gas on the Morphology and Confluence of Human Primary Gingival Keratinocytes—PGK

In the case of primary gingival keratinocytes, the results were slightly different after 20 s of exposure to ozone compared to those obtained for fibroblasts, such as minor changes in the shape of keratinocytes and an increase in confluence (see Figure 4).

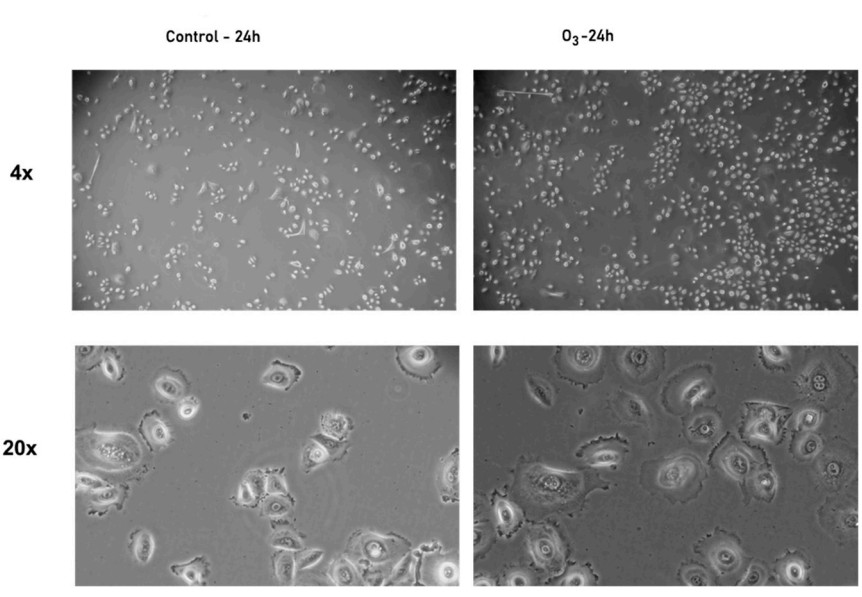

PGK – primary human gingival keratinocytes – 20s ozone therapy

Control – 24h          O₃–24h

4x

20x

**Figure 4.** Microscopical aspect of the primary gingival keratinocytes—PGK at 24 h post-exposure to ozone gas for 20 s. Pictures were taken with 4× and 20× lenses.

Since no significant changes in cells morphology and confluence were observed in the case of gingival keratinocytes, we checked whether exposure to ozone gas for longer periods of time, such as 40 s and 60 s, might induce changes in morphology (Figures 5 and 6).

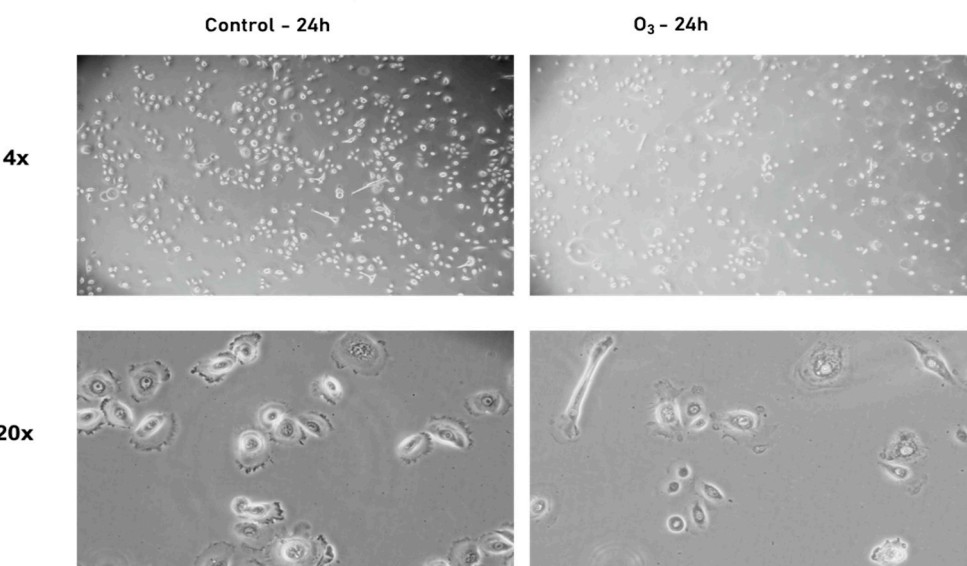

**Figure 5.** Microscopical aspect of the primary gingival keratinocytes—PGK at 24 h post-exposure to ozone gas for 40 s. Pictures were taken with 4× and 20× lenses.

**Figure 6.** Microscopical aspect of the primary gingival keratinocytes—PGK at 24 h post-exposure to ozone gas for 60 s. Pictures were taken with 4× and 20× lenses.

As can be seen in Figures 5 and 6, exposure to gaseous ozone for longer periods—40 and 60 s—led to a decrease in the confluence of keratinocytes and even the presence of cells floating in the culture medium. Some changes in morphology similar to those seen in fibroblasts were observed, but were not as intense.

These results indicate that fibroblasts have a higher sensitivity to gaseous ozone application compared to keratinocytes. This effect cannot be defined as toxic, because no dead cells were observed, only a change in their morphology, which was reversible after 48 h post-exposure.

## 4. Discussion

According to Hodson et al., 2007, dental surfaces subjected to ozone treatment become more resistant to acid attack and the appearance of a carious lesion [38].

The most important aspect of the present study is that, following ozone gas therapy, the degree of demineralization of the tooth structure, measured by laser fluorescence, decreased, with the establishment of an immediate dental remineralization process in all the dental units studied. For all dental units, there was a registered decrease from 14.27 initial mean value of mineralization to 7.90 immediately after ozonation.

According to Duggal et al., 2012, there are no significant differences between the use of gaseous ozone and fluorine in high concentrations for the inhibition of enamel and dentine demineralization [39].

Another important aspect of the measurements is that the ozone-initiated mineralization was a long-lasting process, which continued beyond the one-month interval after the therapy. In measurements carried out one month after the O3 therapy, there was a decrease in mineralization mean values from 7.90 to 6.13.

Following ozone prevention therapy, we succeeded in improving the quality of dental hard tissue from the initial values of 13–24, measured by laser fluorescence, which is equivalent to a dental tissue with an initial demineralization of enamel with a medium risk of caries formation, to the range 0–12, which corresponds to a healthy, hard dental substance with a minimal risk of caries formation.

After the initial measurements by laser fluorescence, the obtained values were higher in lower molars compared to the upper ones, so the degree of demineralization in teeth 3.6 and 4.6 was higher than in teeth 1.6 and 2.6. These higher values for the lower teeth are most likely the result of a higher lingering of bacterial plaque and food debris in the lower arch, which in turn leads to a demineralization in the occlusal surface of molars 3.6 and 4.6, but without significant differences between them.

Another notable aspect shown by the measurements immediately after ozonation is that, the higher the demineralization in dental units, the higher the efficiency of the remineralization.

In addition, the present study analyzed the effects of ozone gas therapy on the morphology and confluence of both primary gingival fibroblasts—HGF—and keratinocytes—PGK. Exposure times were decided taking into account previous studies investigating the effects of ozone on tooth surfaces. Other studies [40] showed that exposure of tooth surfaces to ozone for 20 s leads to the destruction of 99.9% of cariogenic microorganisms, having a preventive effect. An exposure of 40–60 s has been proven to significantly reduce the numbers of *Streptococcus mutans*, one of the determining microorganisms involved in the appearance of carious lesions [41]. Gas ozone application for 60–120 s on fractured teeth, followed by a long-term temporary filling, can improve treatment outcomes [42]. In the case of fibroblasts, exposure to a single dose of ozone gas for 20 s induced significant changes both at 24 and 48 h after exposure. Due to the changes in the fibroblasts' morphology and confluence that were observed after the 20 s exposure, an experiment with longer periods (40 and 60 s) of ozone exposure was not carried out.

Gingival keratinocytes were less sensitive to gaseous ozone exposure when compared to fibroblasts. Only after a 60 s exposure of the keratinocytes to the ozone therapy were there significant morphological changes (Figure 6) similar to the ones detected in fibroblasts exposed for 20 s (Figure 3).

The results obtained by exposing the specific cell lines of gingival tissue, fibroblasts, and keratinocytes to different time intervals of gas ozone therapy show different changes in the cells depending on their type. Among the studied cells, we can say that the gingival fibroblasts showed a marked sensitivity even from the lowest dose of ozone; contrarily, in the case of keratinocytes, structural changes were observed at higher doses.

In a study developed by Borges and collaborators, it was demonstrated that ozonated water at a concentration of 8 μg/mL had no cytotoxic effect on keratinocytes and oral fibroblasts; moreover, a slight increase in cell density was observed compared to control cells. In

contrast, in the present study, the exposure of cells to ozone in gaseous form demonstrated a sensitivity of the two types of cell lines, which denotes different manifestations of cells depending on how ozone is administered [43].

The results obtained are consistent with those reported by other authors in specialized studies. The group of Huth compared the cytotoxicity of chlorhexidine, NaOCl (5.25%, 2.25%), and $H_2O_2$ (3%) to gaseous ozone, and their results showed a slight cytotoxic effect of gaseous ozone, which was lower than that of chlorhexidine, NaOCl, and $H_2O_2$. Similar data emerged from the current study highlighting the cytotoxic effect of ozone in gaseous form on fibroblasts and keratinocytes [18].

In another in vitro study performed by Nam-Kyoung, keratinocytes were tested with different concentrations of ozone gas (between 1 ppm and 3 ppm) over a period of 1 min, and morphological changes were observed at lower doses, whereas at increased doses, specific signs of apoptosis as DNA fragmentation and cell membrane damage occurred. Both the present study and the one conducted by Nam-Kyoung show damage of keratinocytes with increasing exposure to ozone; in Korea, by increasing the concentration but maintaining the exposure time, and in the current study, by increasing the exposure time and maintaining the concentration [44].

## 5. Conclusions

Biological ozone therapy can be used as a common measure to prevent tooth decay, to take the tooth out from the risk area, to reduce bacterial plaque from pits and fissure, and to create the premises of the enamel remineralization process. The mode of application of ozone use is easy and non-invasive, with no major contraindications, and was accepted by all patients included in the study. Following the bio-oxidative ozone therapy, an improvement in enamel quality with significant changes in the demineralization values of dental tissue was found, from equivalent initial enamel demineralization values to values that corresponded to those of integral enamel, without the risk of a carious process. The remineralization process was initiated immediately, but continued for the next 30 days, and the higher the demineralization in the dental units, the higher the efficiency of the remineralization.

Taking into consideration the wide use of ozone as therapy for a myriad of dental pathologies, it became mandatory to evaluate the potential toxic impact of ozone on the resident cells (keratinocytes and fibroblasts) of the gingival tissue that come in close contact with this agent. The present study showed that exposure to 20, 40, and 60 s of gaseous ozone determined a cell-type-dependent response, as follows: the gingival keratinocytes (that form the outer layer of the gingiva) were affected (changes in cells morphology and a decreased confluence) by ozone after only 60 s exposure, whereas in the case of gingival fibroblasts, the morphological changes were observed at the shortest interval of exposure—20 s. These data highlight an increased susceptibility of gingival fibroblasts to gaseous ozone toxicity as compared to keratinocytes, and could be used as background for further studies concerning gaseous ozone's noxious effects.

**Author Contributions:** Conceptualization, A.G., C.A.D., A.D.F. and O.I.B.; methodology, A.D.S., A.D.F., D.C. (Doina Chioran and Dorina Coricovac); software, A.D.S., V.T.A., R.S.R.; validation, A.G., D.C. (Doina Chioran and Dorina Coricovac) and R.B.; formal analysis, I.P., D.C. (Doina Chioran and Dorina Coricovac), C.A.D.; investigation, A.G., D.J., A.D.F.; resources, C.A.D., A.G.; writing—original draft preparation, A.D.F., A.D.S., O.I.B., R.S.R., R.C.R.; writing—review and editing, A.D.S., I.P., C.A.D.; visualization, V.T.A., I.P., R.S.R., L.-C.R.; funding acquisition, D.C. (Doina Chioran and Dorina Coricovac), I.P., O.I.B. All authors have read and agreed to the published version of the manuscript.

**Funding:** This research was financed by "Victor Babes" University of Medicine and Pharmacy, Eftimie Murgu Square, No. 2, 300041 Timisoara, Romania.

**Institutional Review Board Statement:** The study was conducted according to the guidelines of the Declaration of Helsinki, and approved by the Commission of Ethics for Scientific Research of the "Victor Babes" University of Medicine and Pharmacy (No. Av_08_ScurtuA_MD_19).

**Informed Consent Statement:** Informed consent was obtained from all subjects involved in the study.

**Data Availability Statement:** The data used to support the findings of this study are included within the article.

**Conflicts of Interest:** The authors declare no conflict of interest.

## Abbreviations

| | |
|---|---|
| O$_3$ | Ozone |
| SARS -CoV-2 | Severe acute respiratory syndrome coronavirus 2 |
| GSH | Glutathione |
| NICE | National Institute of Clinical Excellence |
| M1 | Permanent molar |
| ATCC | American Type Culture Collection |
| PGK | Primary gingival keratinocytes |
| HGF | Primary gingival fibroblast |
| DNA | Deoxyribonucleic acid |
| FGF | Fibroblast Growth Factor |

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
