# Peer review of "The Biological Effects of Ozone Gas on Soft and Hard Dental Tissues and the Impact on Human Gingival Fibroblasts and Gingival Keratinocytes"

_processes, doi:10.3390/pr9111978_

Round 1
Reviewer 1 Report
No comments and suggestions for authors.
Author Response
Thank you very much for the time allocated for the review of the MS- “ The biological effects of ozone gas on soft and hard dental tissues and the impact on human gingival fibroblasts and gingival keratinocytes”.
Best regards,
Alexandra Scurtu,
Timisoara,
25th of October, 2021

Reviewer 2 Report
The methodology for conducting the research itself is carried out correctly. This is not an extensive scientific problem, but it is an interesting voice in the discussion on the use of ozone in dental therapy.
My doubts are whether the authors controlled the parameters of the ozone generator. Why the research does not take into account the concentration of ozone generated (we do not know any parameter related to it). It also seems that the gas flow rate should also be analyzed. It cannot be that in a scientific study only one setting, the so-called factory devices - especially a specific one. If the device does not allow this parameter to be changed, different devices should be used, or a test stand should be developed that will enable these parameters to be changed. In general, there is an abundance of ozone generators on the market and it is possible to perform such tests without using a specific device. Anyway, the parameters of the ozone stream should be precisely described, including, in particular, its concentration. Without it, the article is only suitable for the manufacturer's brochure. There should also be a subsection critically assessing these parameters, i.e. in particular whether increasing or decreasing these parameters will change the response of these parameters. In particular, work on cell colonies should contain such information.
Authors in general tend to uncritically accept data from specific devices. Units are entered from the device without knowing what they mean or how they were calculated.
Please rewrite the text so that there is specific information about the process parameters and include a critical evaluation of the change of these parameters for clinical success.
At the same time, I would like to point out that trade names should not be abused in such text, which is common here.
At the same time, the authors basically confirm that such studies have already been performed by other researchers. If this is true, their cognitive values are minimal. Please take an opinion on this.
Author Response
Thank you very much for the time allocated for the review of the MS- “ The biological effects of ozone gas on soft and hard dental tissues and the impact on human gingival fibroblasts and gingival keratinocytes” as well as for your pertinent observations.
Please see below a point-by-point response to your comments:
1. My doubts are whether the authors controlled the parameters of the ozone generator. Why the research does not take into account the concentration of ozone generated (we do not know any parameter related to it). It also seems that the gas flow rate should also be analyzed. It cannot be that in a scientific study only one setting, the so-called factory devices - especially a specific one. If the device does not allow this parameter to be changed, different devices should be used, or a test stand should be developed that will enable these parameters to be changed. In general, there is an abundance of ozone generators on the market and it is possible to perform such tests without using a specific device. Anyway, the parameters of the ozone stream should be precisely described, including, in particular, its concentration. Without it, the article is only suitable for the manufacturer's brochure. There should also be a subsection critically assessing these parameters, i.e. in particular whether increasing or decreasing these parameters will change the response of these parameters. In particular, work on cell colonies should contain such information.
Response: We appreciate reviewer’s comment and we have introduced the device parameters.
2. Authors in general tend to uncritically accept data from specific devices. Units are entered from the device without knowing what they mean or how they were calculated.
Response: Thank you for your comment. Explanations regarding the use of different cut-off values have been added in the Material and Methods section to clarify why they were chosen.
3. Please rewrite the text so that there is specific information about the process parameters and include a critical evaluation of the change of these parameters for clinical success.
Response: Thank you very much for your comment. The text has been rewritten with specific information about the process parameters.
4. At the same time, I would like to point out that trade names should not be abused in such text, which is common here
Response: Thank you for your valuable suggestion. The text has been rewritten using the appropriate procedures.
5. At the same time, the authors basically confirm that such studies have already been performed by other researchers. If this is true, their cognitive values are minimal. Please take an opinion on this.
Response: Thank you for your comment. It is true that several studies have been performed to study the effect of ozone on dental tissues. The important aspect of the measurements is that the ozone-initiated mineralisation has been a long-lasting process, which continued beyond the one-month interval after the therapy. In addition, the present study analysed the possible harmful effects of ozone gas therapy on the morphology and confluence of both primary gingival fibroblasts - HGF and keratinocytes – PGK for different time intervals. In the case of fibroblasts, exposure to a single dose of ozone gas for 20s induced significant changes both at 24 and 48h after exposure. Gingival keratinocytes were less sensitive to gaseous ozone exposure as compared to fibroblasts. These data highlight an increased susceptibility of gingival fibroblasts of gaseous ozone toxicity as compared to keratinocytes, and could be used as background for further studies concerning gaseous ozone noxious effects.
We hope you find the revised manuscript acceptable for publication. Thank you once again for your consideration.
Best regards,
Alexandra Scurtu,
Timisoara,
25th of October, 2021

Round 2
Reviewer 2 Report
the changes made are favorable to the final text. There was information about the gas flow and the time of its use. However, there is no discussion regarding the selection of these parameters, which I wrote about earlier. The researcher should not be guided by the standard settings proposed by the manufacturer. Please complete the text with such information and information on how they were selected.
Author Response
Thank you very much for the time allocated for the review of the MS- “ The biological effects of ozone gas on soft and hard dental tissues and the impact on human gingival fibroblasts and gingival keratinocytes” as well as for your pertinent observations.
Please see below a point-by-point response to your comments:
1. the changes made are favorable to the final text. There was information about the gas flow and the time of its use. However, there is no discussion regarding the selection of these parameters, which I wrote about earlier. The researcher should not be guided by the standard settings proposed by the manufacturer. Please complete the text with such information and information on how they were selected.
Response: Thank you for the favourable comments. Time for the exposure has been selected according to exposure needed during dental treatments. The authors wanted to assess possible side effects of ozone on the morphology and confluence of human primary gingival cells: keratinocytes (PGK) and fibroblasts (HGF), cells that represent most of the oral cavity resident cells. Times of exposure have been decided taking into account previous studies in correlation to the effect of ozone on tooth surfaces. Other showed that exposure of tooth sur-faces to ozone for 20 seconds leads to the destruction of 99.9% of cariogenic microorganism, being able to have a preventive effect. An exposure of 40-60 seconds has been proven to significantly reduce the numbers of Streptococcus Mutans, one of the determining microorganisms involved in the appearance of carious lesions. Gas ozone application for 60-120 seconds on fractured teeth, followed by a long term temporary filling can improve the treatment outcomes.
We hope you find the revised manuscript acceptable for publication. Thank you once again for your consideration.
Best regards,
Alexandra Scurtu,
31th of October, 2021